# Cysteine–Silver–Polymer Systems for the Preparation of Hydrogels and Films with Potential Applications in Regenerative Medicine

**DOI:** 10.3390/gels9120924

**Published:** 2023-11-23

**Authors:** Dmitry V. Vishnevetskii, Arif R. Mekhtiev, Dmitry V. Averkin, Elizaveta E. Polyakova

**Affiliations:** 1Department of Physical Chemistry, Tver State University, Building 33, Zhelyabova Str., Tver 170100, Russia; elizabeth03pol@gmail.com; 2Institute of Biomedical Chemistry, 10 Building 8, Pogodinskaya Str., Moscow 191121, Russia; 3Russian Metrological Institute of Technical Physics and Radio Engineering, Worker’s Settlement Mendeleevo, Building 11, Moscow 141570, Russia; averkindmitry@gmail.com

**Keywords:** L-cysteine, silver nanoparticles, biocompatible polymers, self-assembly, hydrogels, microstructure, porous films, cytotoxicity

## Abstract

Herein, the problem concerning the poorer mechanical properties of gels based on low molecular weight gelators (LMWGs)—L-cysteine and silver nitrate—was solved by the addition of various polymers—polyvinyl alcohol (PVA), polyvinyl pyrrolidone (PVP) and polyethylene glycol (PEG)—to the initial cysteine–silver sol (CSS). The physicochemical methods of analysis—viscosimetry, UV spectroscopy, DLS, and SEM—identified that cysteine–silver hydrogels (CSG) based on PVA possess the best rheological properties and porous microstructure (the average pore size is 2–10 µm) compared to gels without the polymer or with PVP or PEG. Such gels are able to form cysteine–silver cryogels (CSC) and then porous cysteine–silver films (CSF) with an average pore size of 10–20 µm and good mechanical, swelling, and adhesion to skin characteristics as long as the structure of CSS particles remains stable. In vitro experiments have shown that hydrogels are non-toxic to normal human fibroblast cells. The obtained materials could potentially be applied to regenerative medicine.

## 1. Introduction

There are different biomaterials and approaches to their creation for use in regenerative medicine [1]. They can be divided into the following two types: natural [2] and synthetic [3]. For instance, collagen, laminin, fibronectin, and silk are naturally occurring materials, whereas manufactured polymers, chemicals, metals, or other synthetically derived substrates are synthetic. Biomaterials of both types have distinct advantages and disadvantages depending on the requested requirements. Along with them, polymeric hydrogels are one of the most promising objects since they can be injectable and compatible with a variety of fabrication technologies, including 3D printing, micropatterning, and electrospinning and, thus, expand the clinical applicability of biomaterials [4].

In recent times, hydrogels based on low molecular weight gelators (LMWGs) have started attracting increased attention, including the synthesis of materials possible for use in regenerative medicine [5,6]. Some of them possess enhanced biological properties; their formation proceeds in terms of the self-assembly process, which is characteristic of such systems, caused by the weak reversible interactions (hydrogen bonding, π–π stacking, Coulomb forces, etc.) of initial molecules and, depending on the chemical nature of precursors, such gels can be of a thixotropic nature [7]. Thus, in most cases, these gels have quite a low viscosity and cannot form films. To solve this problem, LMWGs can be combined with polymers, and there are five key categories of LMWG-polymer combinations [8]: (1) the direct polymerization of self-assembled LMWG fibers through polymerizable groups in LMWG molecules; (2) embedding the LMWG network in a polymerizable solvent; (3) the addition of a non-gelling polymer in solution to LMWGs; (4) the addition of a polymer in the solution that is capable of directed and controlled interactions with the LMWGs; (5) the combination of LWMGs with a polymer that is also capable of gelation (i.e., gelatin) in order to yield a hybrid gel.

LMWGs based on natural components, e.g., amino acids and their derivatives, have shown potential to be utilized and obtain various biomaterials [9,10,11,12]. Such gels have been known to the scientific world for more than a century [13], but their structure was deciphered only in the late 1990s [14]. Then, it was shown that amino acids containing sulfur (such as L-cysteine (CYS), N-acetyl-L-cysteine (NAC), gluthathione (GH) and some salts of heavy metals—copper, silver, gold) could form mechanically weak thixotropic gels [15,16,17,18,19] with anticancer [20,21], photocatalytic [22] and antibacterial/antibiofilm properties [23]. The key role of gel formation in such systems lies in the interaction of the reactive thiol group of the amino acid with metal ions. The gel can be formed in one stage [20,24] or through the sol stage [19,23], depending on the chemical nature of the amino acid and silver salt. The sol–gel transition is initiated by the addition of low-molecular-weight electrolytes (e.g., sulfates and chlorides of metals) [25]. We recently studied in detail the structure of particles from which CYS– and NAC–silver-based sols and gels were formed and showed its effect on the cytotoxicity and bioactive activity of the final materials [23,26]. The self-assembly process in these systems leads to the formation of particles consisting of silver nanoparticles (AgNPs) or silver nanoclusters (AgNCs) in the form of “core” and CYS/Ag^+^ or NAC/Ag^+^ complexes as their “shell”. Carboxylic and amino groups are located on the surface of particles and are responsible for the colloidal stability of the systems and their ability to form a gel.

In the present work, we report the solution to the problem connected with the poor mechanical properties of cysteine–silver gels (CSG) via the addition of aqueous solutions of biocompatible and commercially available polymers such as polyvinyl alcohol (PVA), polyvinyl pyrrolidone (PVP) and polyethylene glycol (PEG). This approach corresponds to the fifth category mentioned above. It was shown that polymers have good compatibility with cysteine–silver sol (CSS), and the resulting solutions are able to form gels by means of adding sodium sulfate. Gels based on PVA are the most mechanically strong and stable due to their ability to form hydrogen bonds with the particles of CSS. These simple manipulations with CSS/PVA-based gels lead to the obtaining of porous films in which the structure of CSS particles remains stable. These films have good mechanical, swelling, and adhesion-to-skin properties. In vitro experiments have shown that normal human fibroblast cells tend to slightly proliferate with time when mixed with gels. Thus, novel materials based on LMWGs with enhanced properties could find potential applications in regenerative medicine.

## 2. Results and Discussion

### 2.1. Characterization of Gels

According to Figure 1, one can see the gradual change in material properties after the addition of the polymer to the initial cysteine–silver sol (CSS). In order to find out in detail the processes occurring between the CSS and polymers, as well as the nature of intermolecular interactions between these objects, modern physicochemical methods of analysis were used. The composition of CSG is given in Table 1. Rheological tests of the obtained hydrogels (CSG) are shown in Figure 2a. The kinetic curves (Figure 2a, top) gradually reached a permanent value; here, the polymer addition increased the viscosity of CSG compared to the gel without the polymer. CSG/PVP and CSG/PEG viscosities varied slightly depending on the concentration and molecular weight of the polymers compared to the control sample (Figure 2a, bottom). However, the viscosity of CSG/PVA increases by seven times. The higher the PVA content in the system, the higher its viscosity. The molecular weight of PVA had a slight effect on the gel viscosity. Thus, macromolecules interact differently with the CSS. Similar dependencies were obtained in [27,28] by studying the rheological properties of highly charged mixed micelles of potassium oleate/N-octyltrimethylammonium bromide after adding the PVA. Particles in CSS also had a surface charge [23]. PVA, PVP, and PEG are neutral-charged polymers, and it seems the nature of CSS interactions with them is non-ionic.

The UV spectra of the CSG are presented in Figure 2b. A similar picture is observed for all samples (Figure 2b, top): the position of the absorption bands at 315 and 390 nm [2], which is attributed to the formation of L-cysteine/Ag^+^ complexes and the local surface plasmon resonance (LSPR) of AgNPs [23,26], respectively, does not change as the polymer is added to the CSS. Therefore, macromolecules do not destroy the CSS particle’s structure. In the case of CSG/PVA, the absorbance at 315 nm has a sharp jump, and then a smooth change takes place with an increase in the polymer concentration compared with the control (Figure 2b, bottom). By contrast, for the CSG/PVP and CSG/PEG, the absorbance drops sharply, especially for the hydrogel with PEG. It is probably because the PVA acts as a linker between CSS particles, while at the same time, PVP and PEG seem to destabilize the gel network like chaotropic agents [29]. pH values do not change for CSG/PVP and CSG/PEG. The pH of the gel without the polymer was 2.55. pH shifted to 2.9. and 3.1 for CSG/PVA (50 and 100 kDa, respectively). In work [30], it was shown that the ratio of deprotonated and protonated carboxyl groups that were located on the surface of particles [26] in CSS was 4:1. Thus, there was a possibility of hydrogen bonds forming between the CSS particles and PVA. The particle size distribution is a bimodal for all systems (Figure 2c, top). The addition of the polymer shifts the peaks of slow and fast modes to the region of large particle sizes and leads to a decrease in the PDI, which indicates the process of structuring. Particle sizes grow slightly with increasing polymer concentrations for CSG/PVP and CSG/PEG (Figure 2c, bottom). For CSG/PVA, the particle size increases by 2–4 times, both in slow and fast modes. The greater the molecular weight of the polymer, the larger the size of the particles. It should be noted that the size of particles in CSS (fast mode) is the same as the size of polymer macromolecules. Consequently, the formed aggregates are probably CSS–polymer complexes. Particles in CSS possess a surface charge, and their value is positive (the zeta potential is +60 mV for CSS and +40 for CSS/Na_2_SO_4_) [23]. If the assumption above about the nature of interactions between CSS and macromolecules is correct, then we should observe the constant value of the zeta potential. Indeed, its value practically does not depend on the chemical nature, molecular weight, or concentration of the polymer (Figure 2d, top). One can see that there is a sharp drop in electrical conductivity, especially for CSG/PVA. Here, conductivity is determined mainly by the following two factors: the concentration of active particles and the speed of their movement. In accordance with the above data, the concentration of absorbing particles decreases, and the particle size practically does not change for CSG/PVP and CSG/PEG. The vice versa picture is observed for CSG/PVA as follows: due to the significant increase in particle sizes with the increasing polymer concentration and according to the Einstein–Stokes equation, the coefficient of their diffusion decreases, meaning that consequently the particle velocity drops.

Finally, it is necessary to consider how the studied properties of CSG affect their microstructure (Figure 3). There is a spatial network formation for all CSGs and nothing for CSSs and polymers. The formation of a porous regular structure was observed only for the CSG/PVA with an average pore size of 2–10 µm. The gel network was irregular, looser, and had breaks for CSG/PVP and CSG/PEG. It should be noted that hydrogels in their native state contain large amounts of water, but experiments were performed after vacuum drying, and this stage could influence the final microstructure of the samples. Certainly, the best way to study the morphology of such hydrogels is the application of CryoSEM or CryoTEM technologies, but they are quite unaffordable. In the present case, when hydrogels were dried at ambient conditions (r.t.), the observed microstructure of samples with various added polymers was the same, and it looked like the initial CSS. It seems that when the water evaporates slowly, the resulting gel microstructure collapses.

### 2.2. Characterization of Films

Biomaterials used in regenerative medicine should have biocompatibility since they influence the development of new tissue to a great extent. Such materials should be non-toxic and non-injurious with respect to living human tissue, blood cells, and the immune system [31]. It is well known that PVA is an affordable, large-tonnage biocompatible polymer that is used, for instance, as a blood plasma substitute. The conventional preparation of PVA hydrogels consists of the addition of various chemical or thermal cross-linking agents such as glutaraldehyde [32], sulfosuccinic acid [33], succinic acid [34], glyoxal [35], dianhydrides [36], maleic acid [37], sodium hexametaphosphate [38], citric acid [39], trimetaphosphate [38], and formaldehyde [40]. However, the presence of these substances in the final polymer structure can cause different harmful side reactions. Furthermore, the interactions of drugs encapsulated in a polymeric matrix with cross-linking agents can give rise to other toxic by-products [41]. The solution to this problem is to use the freeze/thaw process, which proceeds without adding any cross-linking agents [42]. Thus, here, the process of film preparation consisted of the stage of CSG freezing. Only CSG/PVA could form the cryogel mass (CSC) and then films (CSF) (Figure 1). CSG/PVP and CSG/PEG have the possibility of forming the film, but only via conventional pouring and drying. But such films quickly dissolve after contacting water. Here, CSF, based on PVA (2 wt. %), swells in water, has good mechanical properties, and has adhesion to the skin’s surface. PVA cryogels and cryostructurates are well-known objects discovered by Prof. Lozinsky [43]. Figure 4a (top) shows that when moving from CSG to CSC and CSF, the structure of CSS particles remains stable. The conversion of CSS to CSF is about 100% (Figure 4a, bottom). CSF/PVA have the same porous structure as CSG/PVA, but, in this case, they are layered (Figure 4b,c). CSF/PVA (50 kDa) possesses a more regular morphology than CSF/PVA (100 kDa). The average pore size is 10–20 µm. The CSS particle distribution on the surface and inside the film is homogeneous (Figure 4d,e). These results are very important because the main requirements for materials to be used in regenerative medicine are a porous structure and the absence of cytotoxicity. It is known, for instance, that the various cell lines of fibroblasts actively divide and grow on scaffolds with pores ranging from 2 to 20 μm [44,45,46].

### 2.3. Toxicity of Materials

As mentioned above, the biocompatibility of materials used in regenerative medicine is the main sticking point. It is known that silver nanoparticles and silver ions are potentially toxic not only to bacteria but also to normal human cells [47,48]. This effect can be associated with various reasons. Firstly, the dose-dependent mechanism of toxicity deals with the accumulation of AgNPs or Ag^+^, which is released through oxidative dissolution and plays one of the key roles [47,49]. Secondly, the size of AgNPs significantly influences their cytotoxicity [50,51,52,53]. Finally, AgNPs with irregular surface and surface defects can induce stronger cytotoxicity [54]. Figure 5 shows the toxicity of CSG/PVA to normal human fibroblast cells. One can see that the quantity of cells slightly increases with the increase in incubation time. Despite the fact that materials used in regenerative medicine must meet the requirement for intensive cell growth, our experiments were performed with lung fibroblasts, which are more sensitive to cytotoxic influence in comparison with dermal ones. Furthermore, we have previously shown that nanosilver-containing sols and hydrogels prepared using L-cysteine or N-acetyl-L-cysteine are also non-toxic to blood cells, erythrocytes, and immune system cell macrophages [23]. Thus, novel hydrogels can be used as a system for the cultivation of normal cells and, most likely, in the 3D printing of the finished porous matrix with the desired shape and thickness. Here, CSG/PVA and CSF/PVA both have potential for application in regenerative medicine. Indeed, when applying such films as a scaffold in tissue regeneration, for the indirect cell–cell interaction across the scaffold, the pore size should be large enough to ensure cellular nutrition but not too large to prevent cell migration. On the other hand, for the transmigration of transplanted cells out of the scaffold to damaged tissue, the ratio between the cell and pore size is also an important factor [44,45,46]. If we want to use these films directly for wound healing, the porous structure is also relevant because, for active regeneration, oxygen must be available to the wound, and exudate removal must also occur. In addition, the presence of silver nanoparticles in the film structure provides an antibacterial effect. 

## 3. Conclusions

In conclusion, new hydrogels based on the cysteine–silver sol and water-soluble biocompatible polymers have been obtained via the simple mixing of their initial aqueous solutions and the further addition of sodium sulfate. The chemical nature of the polymer plays a key role in the formation of gels with improved rheological characteristics and microstructure, which is characteristic of PVA but not of PVP and PEG. This effect seems to be connected with the hydrogen bonding formation between CSS particles and PVA macromolecules. Further manipulations with CSG/PVA led to the formation of porous films that swelled in water, had good mechanical properties, and adhered to the skin. In vitro cytotoxicity analysis showed that hydrogels are non-toxic to normal human fibroblast cells. The quantity of cells slightly grows at its incubation, with CSG/PVA increasing at the incubation time. The developed approach for the preparation of porous biocompatible gels and films could open new perspectives for their use in regenerative medicine. Our further experiments focus on the estimation of the antibacterial activity of prepared materials, as well as their cytotoxicity in in vivo studies.

## 4. Materials and Methods

### 4.1. Chemicals

L-cysteine (>99%) was supplied by Acros. Silver nitrate (>99%) was obtained from Lancaster. PVA (50 and 100 kDa, 99.25% of hydrolysis), PVP (20 and 40 kDa), and PEG (0.4, 6, and 40 kDa) were purchased from Sigma Aldrich (Saint Louis, MO, USA). All chemicals were used as received. All systems were prepared using de-ionized water.

### 4.2. General Procedure for the Preparation of CSS, CSG, CSC and CSF

The CSS (2 mL) was prepared according to our previous technique [21]: an empty vessel of 0.65 mL was filled with de-ionized water, then 0.6 mL of L-cysteine (CYS, 0.01 M) and 0.75 mL of silver nitrate (0.01 M) were added successively. The ratio of CYS to AgNO_3_ was fixed at 1 to 1.25. The obtained white–yellow opalescent mixture was put on the magnetic stirrer at room temperature (25 °C) for 1 min, and the resulting solution was placed in a dark place for 3 h. As a result, a greenish–yellow transparent sol (CSS) was obtained. CSGs were obtained via the consequential addition of a PVA, PVP, or PEG solution to a CSS (the final concentration of polymer was 0.002, 0.01, 0.02, 1, and 2 wt. %) and Na_2_SO_4_ solution (the final concentration of the electrolyte was 0.0001 wt. %). CSCs were prepared by one cycle of freezing (–20 °C), the thawing (25 °C) of CSG, followed by centrifugation (6 × 10^3^ r.p.m) for 20 min at 25 °C and then water decantation. CSFs were obtained via the vacuum (10^–4^ Pa) drying of CSC on an aluminum foil at 25 °C.

### 4.3. Rheological Test

A vibratory viscometer SV-10 (A&D Company, Tokyo, Japan) was used for the viscosity measurements of the samples. The sensor plate vibration was carried out at a frequency of 30 Hz and a constant amplitude of about 1 mm. In total, 10 mL of the studied systems were prepared in special polycarbonate cups (A&D Company, Tokyo, Japan) for the measurements of viscosity. After 24 h of samples staying in a dark place, these cups were transferred to the viscometer, and the measurements were recorded. The temperature of the experiment was 25 °C.

### 4.4. SEM and EDS

The microstructure and chemical composition of the samples were studied using a raster JEOL 6610 LV electron microscope (JEOL Ltd., Tokyo, Japan) with X-ray system energy dispersive microanalysis Oxford INCA Energy 350. The micromorphology of CSS, CSG, and CSF was studied using the high-vacuum mode with an accelerating voltage of 15 kV. For image acquisition, low-energy secondary electron signals, providing a topographical contrast, and high-energy back-up scattered (reflected) electrons that determined the composition and phase contrast were generated. The elemental chemical composition of the samples was determined via X-ray spectral microanalysis based on the registration and analysis of the energy spectra of the characteristic X-ray radiation excited by electrons passing through the sample. The qualitative and quantitative elemental composition was determined using an energy-dispersive spectrometer (EDS), which sorts photons by their energy. The preparation of samples included spraying the samples on a thin conductive layer of the platinum surface and drying them in a vacuum (10^–4^ Pa). The average platinum coating time was 5 min.

### 4.5. UV Spectroscopy

The electronic spectra of the samples were recorded on the UV spectrophotometer Evolution Array (Thermo Fisher Scientific Inc., Waltham, MA, USA) in a quartz cell with a 1 mm path length.

### 4.6. DLS and Zeta Potential Measurements

The measurement of the intensity of light scattering in the studied samples was carried out using analyzer Zetasizer Nano ZS (Malvern) with a He-Ne laser (633 nm) at a power of 4 mW. For the correct analysis of the particle sizes and zeta potential, after complete gelation, the samples were destroyed via shaking, followed by their dilution at two, four, and eight times. All measurements were performed at 25 °C in the backscattering configuration at an angle of 173°, which provides the highest sensitivity of the device. The results under investigation were mathematically processed via the obtained cross-correlation functions of the diffuse light intensity fluctuations g2, which was carried out in the program Zetasizer Software v7.11, where the solution of the obtained equation of the g2 dependence on the diffusion coefficient was performed using the method of cumulants. The solution to this equation was the function z(D). The hydrodynamic radii of the scattering particles were calculated from the diffusion coefficients using the Stokes–Einstein formula: D = kT/6πηR, where D is the diffusion coefficient, k is the Boltzmann constant, T is the absolute temperature, η is the viscosity of the medium, and R is the radius of the scattering particles. The measurement of the electrophoretic mobility of aggregates in the samples was carried out in U-shaped capillary cuvettes. Zeta-potential distributions were calculated using the Henry equation: UE = 2ezf(Ka)/3Z, where UE—electrophoretic mobility, z—zeta potential, e—dielectric constant, Z—viscosity, and f (Ka)—Henry’s function, f (Ka) = 1.5 for aqueous media.

### 4.7. pH Measurements

The pH of sols and gels, which were destroyed via shaking, was measured using a Seven Multi S70 (Mettler Toledo, Columbus, OH, USA) pH meter.

### 4.8. Cytotoxicity Evaluation of Hydrogels in Wi-38 Cells (MTT-TEST)

A standard human normal embryonic lung cell (Wi-38) obtained from the American Tissues and Cells Collection (ATCC, Manassas, VA, USA, Lot. CCL-75) was kindly provided from the collection of the N.N. Blokhin National Medical Research Center of Oncology of the Ministry of Health of the Russian Federation. These cells were not additionally modified genetically; any new data about this culture were received, and it was only used as a model of normal cells to assess the cytotoxicity of the investigated gels. Cells were adhered to 96-well plates and cultured for 24 h at 37 °C in an atmosphere of 5% CO_2_ in a DMEM medium with the addition of L-glutamine (2 mM), antibiotics (100 units per mL of penicillin and 100 μg/mL of streptomycin) and 10% of FBS. The cells were then incubated in a serum medium with the tested compounds at a sample concentration of 0.3 mM ([Af]) for 24, 48, and 72 h. PBS was added to the culture medium (10μL/well) containing MTT (5 mg/mL), and the cells were incubated at 37 °C for 4 h. Yellow MTT was converted into purple formazan via cellular respiration. The culture medium was removed, and DMSO (100 μL) was added to each well; a plate was stored at 5–8 °C, and optical absorption measurements were carried out the next day in each well at 570 nm on a Multiskan Spectrum Microplate Reader (Thermo Scientific, USA). MTT test readings were averaged from three independent experiments using three independent determinations. The MTT test readings in the absence of test compounds were taken as 100% cell viability.

## Figures and Tables

**Figure 1 gels-09-00924-f001:**
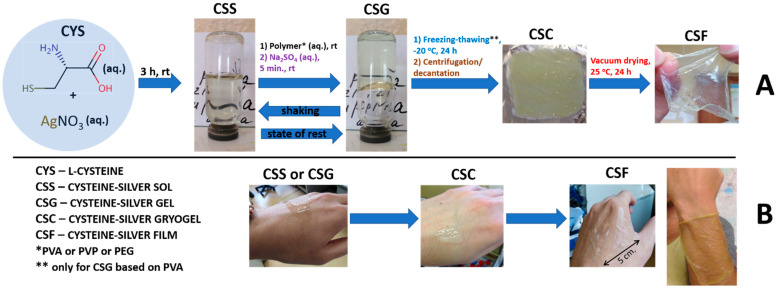
(**A**) The scheme of gels and films preparation. (**B**) Abbreviations and the character of the behavior of the obtained materials on the skin of the hand.

**Figure 2 gels-09-00924-f002:**
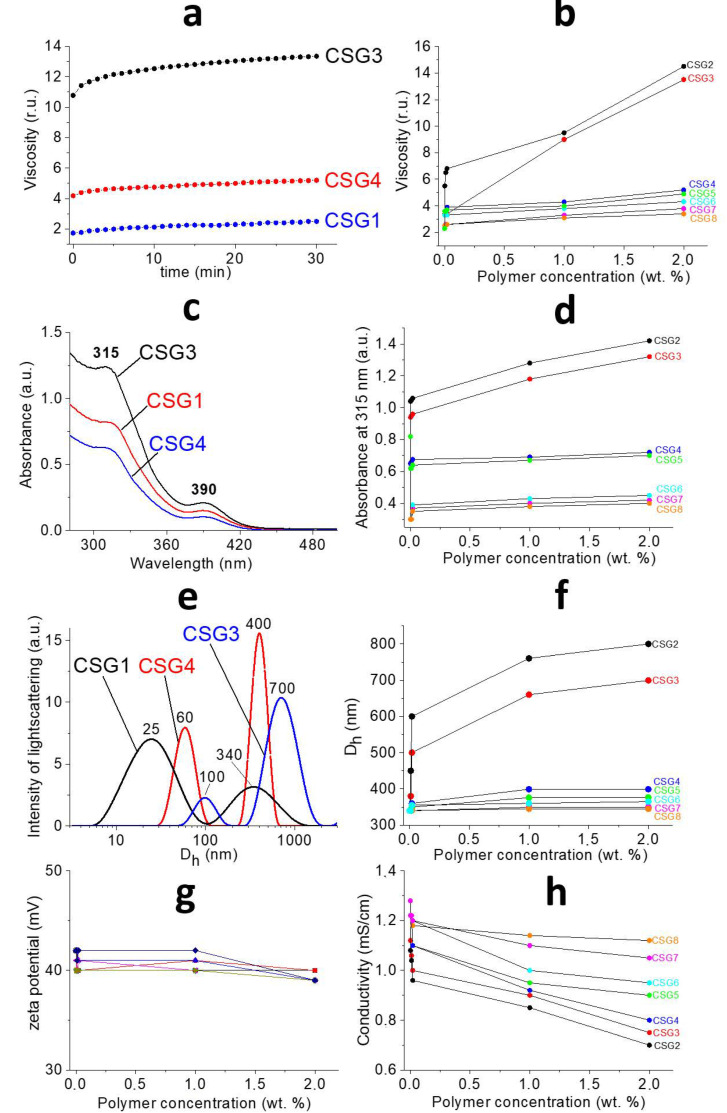
(**a**,**c**,**e**)—The change in viscosity over time, the UV-spectra and particle size distribution of CSG. (**b**,**d**,**f**,**g**,**h**)—The viscosity, A_315_, <D_h_>, ζ and conductivity dependence of CSG from [polymer] and <Mw>. [polymer]—2 wt. %. CSG1—CSG8 see in Table 1.

**Figure 3 gels-09-00924-f003:**
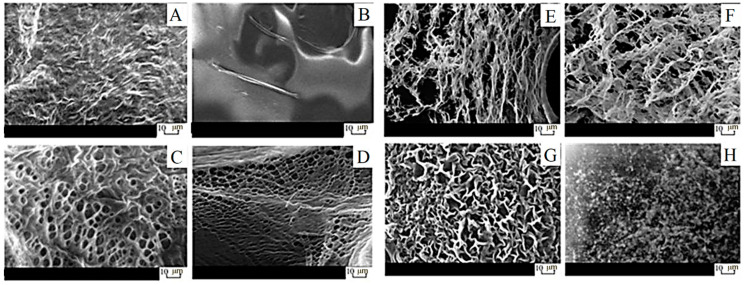
SEM images: (**A**) CSS, (**B**) PVA, (**C**) CSG3, (**D**) CSG2, (**E**) CSG5, (**F**) CSG4, (**G**) CSG6, (**H**) CSG7. [polymer]—2 wt. %. CSG2-CSG7 can be seen in Table 1.

**Figure 4 gels-09-00924-f004:**
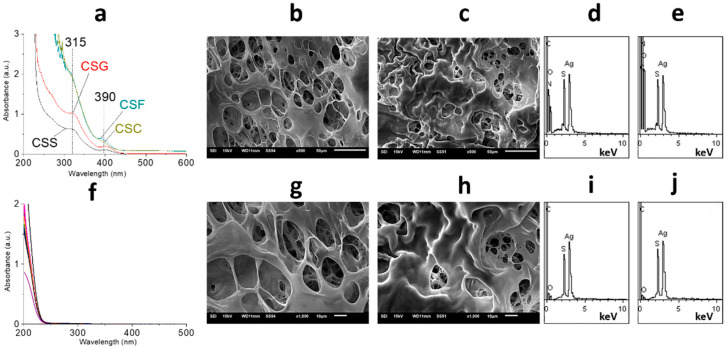
(**a**) UV spectra of materials and (**f**) Supernatants after centrifugation. (**b**,**g**) and (**c**,**h**)—SEM images of CSG3 and CSG2-based films at different scales. (**d**,**i**) and (**e**,**j**) EDS analysis of CSG3 and CSG2-based films on the surface and in the volume. [polymer]—2 wt. %. CSG2, CSG3 can be seen in Table 1.

**Figure 5 gels-09-00924-f005:**
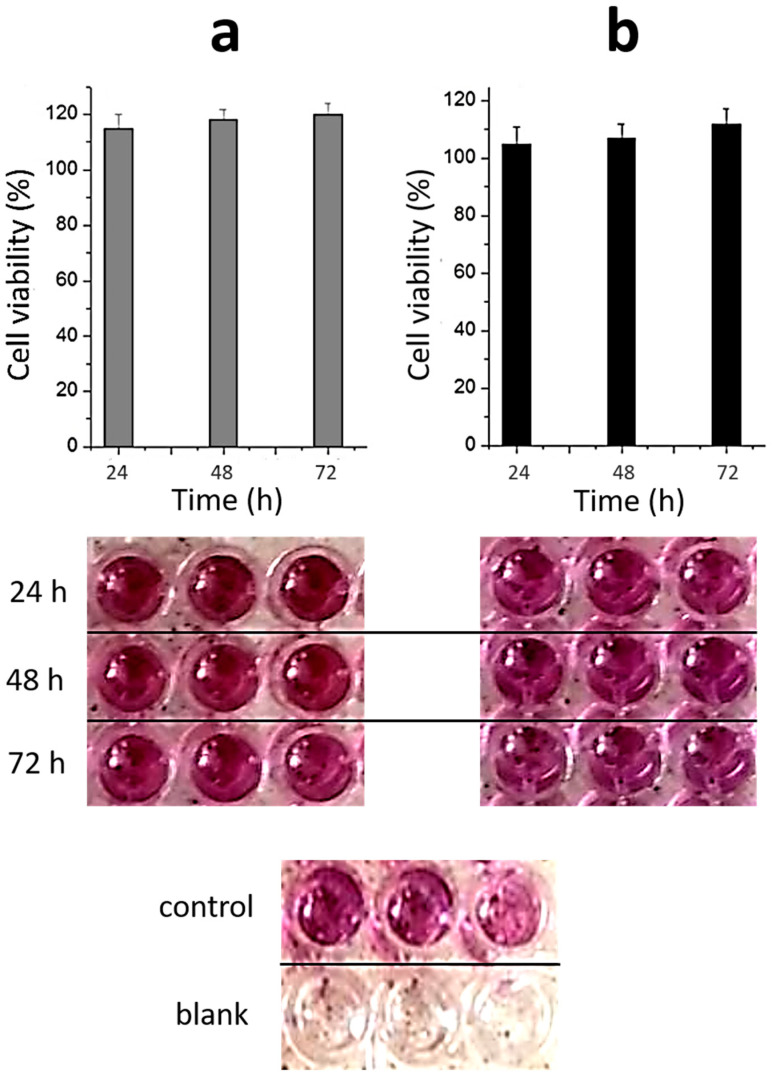
The cytotoxicity of CSG/PVA at different times of incubation with Wi-38 cells. (**a**)—CSG3 and (**b**)—CSG2. [polymer]—2 wt. %. CSG2, CSG3 can be seen in Table 1.

**Table 1 gels-09-00924-t001:** The composition of cysteine–silver–polymer gels.

Entry	Polymer (Molecular Weight kDa)
CSG1 CSG2	No polymer PVA (100 kDa)
CSG3	PVA (50 kDa)
CSG4	PVP (40 kDa)
CSG5	PVP (20 kDa)
CSG6	PEG (40 kDa)
CSG7	PEG (6 kDa)
CSG8	PEG (0.4 kDa)

## Data Availability

All data and materials are available on request from the corresponding author. The data are not publicly available due to ongoing research using a part of the data.

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
