# Peer review of "Cysteine–Silver–Polymer Systems for the Preparation of Hydrogels and Films with Potential Applications in Regenerative Medicine"

_gels, 2023, doi:10.3390/gels9120924_

Round 1
Reviewer 1 Report
Comments and Suggestions for Authors
In this work the authors investigated the effect of combining a series of polymers with supramolecular cysteine-silver hydrogels. Changes in viscosity, porosity, UV absorption, zeta potential and particle size were monitored. An assay on cititoxicity was performed.
It is well known that combining polymers with LMWGs affects the final properties of the hydrogel, hence the manuscript does not report any new concept, in addition it does not present any new element. Moreover, the whole discussion is based on insufficient experimental evidence and a more throrough checking of the manuscript should have been done before submission.
1. In the introduction, in the paragraph dealing with the combinations of polymers with LMWGs specific literature should be included for the five categories mentioned by the authors.
2. In the last paragraph of the introduction, authors should explain to which of the five categories mentioned above belongs the work presented in this manuscript.
3. In general, abbreviations and acronysms should be defined the first time they are used in the text. i.e. PVA, PVP, PEG, CSC, CSF, LSPR
4. A table containing the description and composition of the different samples should be added to the main text
5. Figures 2 and 4 are too small and the captions are very complex. Adding a table with the description of the samples should help to simplify Figure captions
5. Rheological measurements are neccesary to study the mechanical properties, they can give important information about the robustness of the hydrogels
6. I'm wondering if FT-IR analysis could give information about the hydrogen bond interaction proposed by the authors
7. Figure 2d, how did authors get conductivity data?
Comments on the Quality of English Language
Editing for english language is required, both grammar and language need revision. Authors could also use a professional english editing service.
Author Response
Dear referee, thank you very much for such detailed analysis of our manuscript! All corrections are marked in purple color.
You are absolutely right that the article doesn’t report any new concept. But the new element of the article is to use of the well-known concept for the solution of the concrete task. The data obtained in this article has been presented for the first time.
1) We have given the reference on the review article (ref. 8).
2) The sentence has been added.
3,4) Thank you! The table and abbr. have been added.
5) Yes, you are right, we intend to do it. Anyway, rheological properties of the gel without the polymer were obtained earlier (see ref. 19).
6) FTIR analysis is not suitable here since hydrogels have a large amount of water. Certainly, we were removing the water and obtained precipitates, but we didn’t manage to find any big difference in the spectra in depending on the chemical nature of the polymer. Moreover, such approach is not quite right since we remove the water which is responsible for the hydrogen bonds formation between the various molecules. But thank you very much, I have come up the idea to use the Raman spectroscopy for this purpose.
7) The Zetasizer Nano ZS automatically gives the values of the conductivity when the zeta-potential is calculated based on the electrophoretic mobility measurements.

Reviewer 2 Report
Comments and Suggestions for Authors
The research manuscript titled “Cysteine-silver-polymer systems for the preparation of hydrogels and it-based films with potential applications in regenerative medicine”, explains how the Cysteine-silver gel system can be reinforced with various different polymers for improving the property towards applications. The authors have blended the cysteine-silver system with three different polymers and compared them. This article would be good read for researchers working on biomaterials based on metal co-ordination complexes and hydrogel systems for various applications. The authors have carried out appropriate experiments pertaining to the title of the work. However, the following issues have be addressed before accepting for publication.
1. The title is little confusing. “….gels and it-based films with…” A revised and easier to understand title will be better.
2. Line 87, “….according to our previous technique..” reference number is missing.
3. Line 88, revise “0.6 ml of L-cysteine” rather than 0.6 ml.
4. Line 103,104, manufacturer and model number might be sufficient, rather than the webpage.
5. In sample preparation of SEM and EDS, include the platinum coating time.
6. For the DLS and Zeta potential measurements, the authors should include the dilution of their samples. Same applies for the UV spectroscopy.
7. Figure2, the font size of the graphic labels should be increased to improve the text visibility.
8. Line 258-259, grammar check is required.
9. Line 260, cultivation spell check required.
10. Line 258-259, the authors mention that, “One can see that cells proliferation takes place at increasing of the incubation time.”. However, the cell proliferation remains the same at 24, 48 and 72 hours, fig 5 a & B, as the percentage viability is almost the same for all time points. The figure shows that the samples maintain the viability of the cells, however, proliferation of cells over time is doubtful. The authors are advised to revise accordingly.
11. Line 275-276, although it is known that silver ions can exhibit antibacterial properties, however in this study/manuscript, the authors have not conducted any experiments to elicit the antibacterial effect of the hydrogel system that they have prepared. Therefore, it is advised not to mention as antibacterial gels and films, unless the authors perform some antibacterial studies to prove the same.
Comments on the Quality of English LanguageGrammar corrections and spell checks have to be done.
Author Response
Dear referee, thank you very much for such detailed analysis of our manuscript! All corrections are marked in green color.
1) The title has been corrected.
2) The reference has been added.
3) It has been corrected.
4) It has been corrected.
5) The sentence has been added.
6) Concerning DLS and zeta-potential analysis we have noticed it in the experimental part (the second sentence). UV analysis was performed without the dilution.
7) The font size has been increased.
8) The sentence has been corrected.
9) It has been corrected.
10) This sentence and further discussion have been corrected.
11) It has been corrected. Yes, we didn’t have enough time to carry out these experiments and we have just made such assumption based on our article (ref. 23). It will be our further investigations.

Reviewer 3 Report
Comments and Suggestions for Authors
The authors present results of their work on hydrogels with improved mechanical properties. In the title they claim, that the material is designed for the regenerative medicine. However, this statement is not supported in the manuscript.
Major remarks:
In the introduction the authors claim that their material shows good adhesion to skin. I did not find any result supporting such thesis in the manuscript. The only experiment with cells is performed for lung fibroblasts (not dermal cells), moreover it shows only that the materials are not cytotoxic. None experiments showing the adhesive properties is presented. In turn, in paragraph 3.2 the authors report the results on porosity and conclude that their results are very important because the main requirements for materials in the regenerative medicine are porosity and antibacterial activity. The importance of antibacterial activity is raised once more in the conclusions, however no single test presenting the antibacterial effect of the materials is reported in the manuscript.
What is more, based on the cytotoxicity results (Fig.5) the authors conclude that their material have a good potential to be applied in the regenerative medicine. In fact, the presented results show that the number of cells does not decrease after 72h of culture and that proliferation is very limited - the bars in the graph do not change by more than 10%. This means that the proposed materials have rather low application potential in regenerative medicine, where the intensive cell growth should be promoted.
And finally - the SEM images. The hydrogels in the native state contain the large amounts of water. However, SEM experiements are performed in vacuum. What was the drying procedure? What was its effect on the original hydrogel structure?
Comments on the Quality of English LanguageNumerous mistakes, mainly with a/the, 's' ending of verbs, etc.
The larga parts of text look like written in different language and then translated by some AI-based tool.
Author Response
Dear referee, thank you very much for such detailed analysis of our manuscript! All corrections are marked in yellow color.
1) You are absolutely right concerning the statement about the adhesion properties. We didn’t have enough time to do it, but you can see on the Fig. 1 (CSF, bottom) that the final film possesses the good adhesion to the skin, take my word for it, anyway if it is not enough I will delete this statement.
2) Concerning antibacterial properties, I have removed this statement, but we have made such assumption based on our previous results (ref. 23). Actually, if we have porous structure we can add antibacterial drug to this matrix, this is not a problem. Anyway, it will be the next studies.
3) I would say that the percentage of cells still slightly increases. However, this is not enough, you are right. But you have mentioned that our experiments (MTT-test) were performed for lung fibroblasts (not dermal cells). Anyway, it is known that lung fibroblasts are more sensitive to the toxic influence compared to dermal ones. We have used lung fibroblasts as a model of normal human cells.
4) Concerning SEM of hydrogels. Certainly, the best way is to study it via the CryoSEM or CryoTEM, but unfortunately, we didn’t have such possibility. When we dried gels at ambient conditions (r.t.) the observed microstructure of samples with various added polymers was the same and it looked like the initial cysteine-silver sol (Fig. 3.1). It seems when the water evaporates slowly the resulting gel microstructure collapses.

Round 2
Reviewer 1 Report
Comments and Suggestions for Authors
I thank the authors for their responses. The quality of the manuscript has been improved, but the following points should still be addressed.
1. Page 1, lines 38-41. Not all supramolecular gels derived from LMWGs have enhanced biological properties or thixotropy; these are not intrinsic properties of supramolecular gels. They may show the above-mentioned properties, but it depends on their structure. Please correct.
2. Page 1, line 39. In this sentence, I would suggest removing the adjective “unusual” to describe the self-assembly. The self-assembly of LMWGs is well known and well documented in literature.
3. Page 1, line 40. It should be “Coulomb forces” and not “coloumb forces”
4. Page 1, In the introduction, in the paragraph dealing with the combinations of polymers with LMWGs, specific literature should be included for the five categories mentioned by the authors. A review from 2015 is not enough. I guess recent examples can be found in the literature.
5. In the Materials and Methods section, the preparation of gels containing different amounts of polymers (0.002, 0.01, 1 and 2 wt %) is described, however, only gels containing a 2 wt. % are discussed in the manuscript. Authors should remove the other percent from the Materials and Methods section if no data are given, otherwise the data corresponding to these gels should be given as supporting information.
6. Related with point 5, if gels with different wt. % of polymers have been synthesised, then the wt. % should also be indicated in Table 1. Table 1 should also include an entry for the gel without polymer.
7. Please use letters to identify the different plots within a figure, avoid using (top)/(bottom).
8. The captions and Figures 2 and 4 are still very messy. For example, in Figure 2, in some plots line 1 corresponds to CSG2, in others to “gel without the polymer”. Something similar happens with the other lines. To avoid confusion, the authors to complete Table 1 and use that table as a reference. Authors should avoid using numbers to identify the lines within the plots and instead use the acronyms given in the table.
9. The plots in Figure 2 are too small and difficult to read. Please modify the figure so plots are readable.
10. Figure 3. Please, use letters instead of numbers to identify the different images. Use Table 1 as a reference to write the caption of the figure
11. I suggest to add the rheological data to this paper.
12. Regarding the IR, IR of xerogels (dried gels) is commonly used to study hydrogen bonding in other supramolecular gels, for example, in peptide-based supramolecular gels. Shifts of the bands of the functional groups involved in hydrogen bonding can be easily detected. Anyway, I agree with the authors that the use of RAMAN is a good idea as it could provide additional information.
13. Line 31 ‘Both of them have distinct advantages and disadvantages’ Which ones? This sentence is too general.
Comments on the Quality of English Language
Extensive editing of English language required
Author Response
Dear referee, thanks a lot for your comments! All corrections are marked in green color. The manuscript has been extensively edited.
1) The sentence has been corrected.
2) This word has been removed.
3) It has been corrected.
4) I have tried to do my best to find any latest articles but everything leads to this review of 2015.
5) We discuss these data in the main text (Fig.2, the dependence of the various parameters of CSG from the polymer concentration).
6) It has been corrected.
7) The figure has been modified.
8) The figure has been modified.
9) The figure has been modified.
10) The images have been corrected.
11) Yes, I agree with you. But we have used the vibrational viscosimetry which is also suitable for the study of such gels.
12) Thank you. However, in peptide-based gels the molecules form quite strong hydrogen bonds and it reflects in G’’ and G’ modulus which are in 1-2 orders of magnitude higher compared to our gels.
13) The sentence has been corrected.

Reviewer 3 Report
Comments and Suggestions for Authors
Authors responded to some of my remarks.
No further comments.
Author Response
Dear referee, thank you very much for your condescension.
